# Selection of the Depth Controller for the Biomimetic Underwater Vehicle

Michał Przybylski 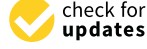

Faculty of Mechanical and Electrical Engineering, Polish Naval Academy, 81-127 Gdynia, Poland;
m.przybylski@amw.gdynia.pl; Tel.: +48-261-262-524

**Abstract:** The aim of this paper is to select a depth controller for innovative biomimetic underwater vehicle drives. In the process of optimizing depth controller settings, two classical controllers were used, i.e., the proportional–integral–derivative (PID) and the sliding mode controllers (SM). The parameters of the regulators' settings were obtained as a result of optimization by three methods of the selected quality indicators in terms of the properties of the control signal. The starting point for the analysis was simulations conducted in the MATLAB environment for the three optimization methods on three types of indicators for three different desired depth values. The article describes the methods and quality indicators in detail. The paper presents the results of the fitness function obtained during the optimization. Moreover, the time courses of the vehicle position relative to the desired depth, the side fin deflection angles, the calculated parameters of the control signals, and the observations and conclusions formulated in the research were presented.

**Keywords:** biomimetic underwater vehicle; depth controller; genetic algorithms; particle swarm optimization; Pareto optimization

## 1. Introduction

In the 21st century, there has been dynamic development of mobile robots. One of the fields is underwater robots, where we can distinguish between ROV (Remote Operated Vehicle) and AUV (Autonomous Underwater Vehicle). Vehicles based on ROV technology, which dates back to the 1960s, have been ideally developed and are successfully used in all kinds of underwater operations [1–3]. A significant limitation of the mobility of these vehicles is the use of cable tether, which leads scientists and engineers to develop AUVs [4]. Autonomy, advanced control and positioning systems allow the realization of many civilian and military tasks. The development of bionics resulted in a new trend in the construction of underwater mobile robots, whose main idea is to imitate underwater animals. These vehicles are called biomimetic underwater vehicles (BUVs) [5,6], which mimic construction and motion kinematics. Most often, the prototype vehicle is inspired by the shape and movement of aquatic creatures, although there are designed based on manta rays, penguins and many others. Nevertheless, it is essential to carefully analyze the animal movement and to develop an appropriate simplified mathematical model, which will be used as close as possible to the BUV [7]. Artificial fins, similar in shape and appearance to the real ones, propel the BUV. Such propulsion is called wave or undulating propulsion, and it can be placed in different parts of the BUV's hull, depending on the type of design, its maneuverability and the speeds it can achieve. An electric motor usually drives it to generate sufficient vehicle thrust through oscillating motion. A sinusoidal function usually describes oscillatory motion, but different parts can be used depending on the type and purpose of the undulating propulsion. To describe oscillatory motion, usually use parameters such as the neutral position, which is the zero position for oscillation, the amplitude of oscillation, which defines the maximum deformation of the fins, and the frequency of oscillation [8].

Control of underwater vehicles has been investigated quite thoroughly. However, still many difficulties occur, among other things, due to environmental disturbance, highly nonlinear behavior of vehicles, the complexity of the vehicle hydrodynamics or the application of new innovative propulsion systems. The articles [9–12] present various systems for controlling the depth of underwater vehicles. They concern both controllers tuned classically and using artificial intelligence algorithms. However, the works presented above concern, to a large extent, ROV or AUV vehicles using a classic screw drive propeller. Biomimetic underwater vehicles can be equipped with an artificial swim bladder [13] similar to a fish bladder, which controls buoyancy and depth. Another solution used to control the depth of biomimetic underwater vehicles [14,15] is changing the locomotion primitives of the fins, usually by changing frequency, amplitude or side fins' phase shift. In more advanced applications, changing the depth can be done by adjusting the angles of attack of the vehicle's control surfaces, their stiffness [16], or surface area of side fins [17] while the vehicle is moving at a certain speed through its wave propulsion.

The main contribution of this paper is to present depth controllers' different tuning methods for innovative BUV wave propulsion drives. The proposed methods use artificial neural networks in the tuning process and the experimental verification system of controller gains to evaluate its performance. In the presented literature, one can notice the lack of use of neural network-based methods for tuning depth controllers based on bioinspired wave propulsion drive. This research could have significant implications for the development of biomimetic autonomous underwater vehicles for various applications such as exploration of the ocean ecosystems, environmental monitoring, or even search and rescue missions. The rest of the paper is organized as follows. The mathematical model of the vehicle is based on the Fossen model [18] with modifications to include the new wave drive used. Commonly used underwater vehicles are driven mainly by a set of screw propellers, whereby the thrust generated by the propellers can be calculated using specific mathematical formulas [19]. In the case of a new wave propulsion system that mimics the action of fish fins, the forces and moments of force acting on the vehicle were calculated using the author's method presented in the literature [20]. Then the applied depth controllers and their control methods are presented. The following section offers the research problem and the results obtained in the simulation process. The final section formulates conclusions and plans for future research.

## 2. Materials and Methods

This chapter presents the simulation model and methods used in the tuning process of the depth controller settings. It contains two main subsections. The first concern describes biomimetic underwater vehicles with an innovative propulsion system. A mathematical model based on the above-mentioned vehicle is described in the further part of this subsection. While the second main subsection deal with the description of controllers, optimization methods and fitness functions used in the simulation process.

### 2.1. Control Object

This subsection describes the construction of a biomimetic underwater vehicle (mini-Cyber Seal) with particular emphasis on the new propulsion system. In the next part, the mathematical model of the vehicle is presented, where special attention is paid to the proposed solutions for modelling the vector of vehicle forces and moments.

#### 2.1.1. Mini CyberSeal

The study used a physical model of the BUV mini CyberSeal vehicle shown in Figure 1, is a smaller prototype of the larger vehicle. The purpose of the downsized BUV was to test the performance of the larger vehicle's counterpart's propulsion system and to test a new type of control. Unlike its predecessors, the mini CyberSeal's propulsion system had two tail fins instead of one. They generate the major thrust and are additionally responsible for changing course. In addition, a side propulsion system is mounted in front of the fuselage,

which generates additional thrust and changes the depth. The rear and side fins are made of polycarbonate and rubber.

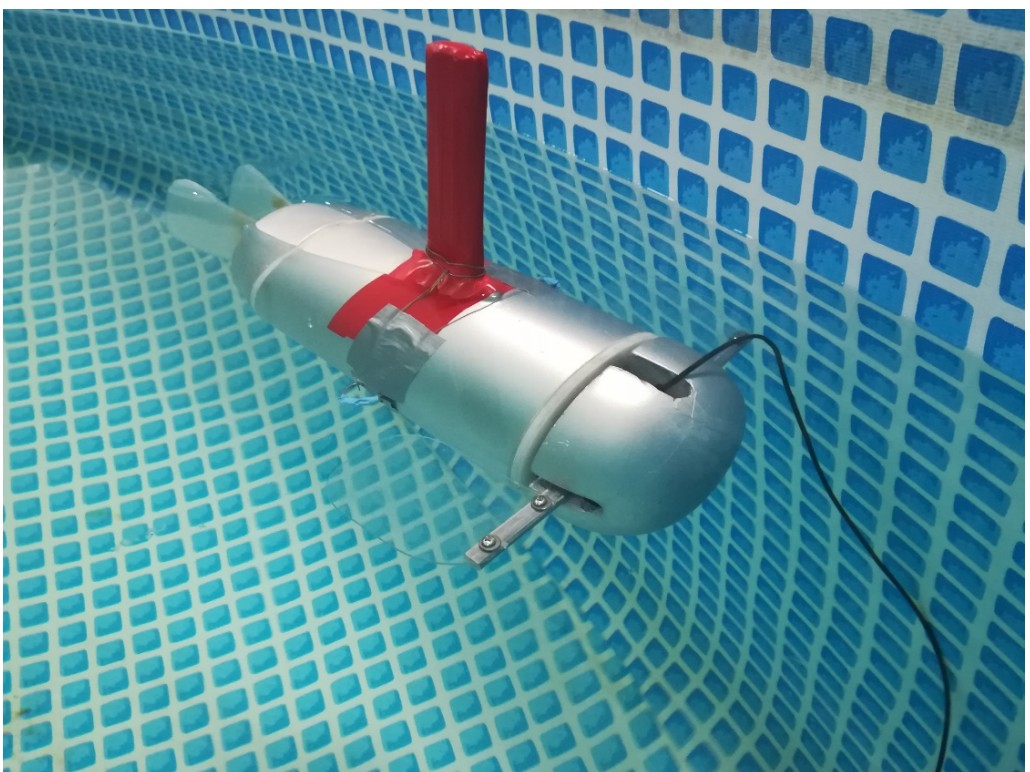

**Figure 1.** Model of the BUV mini CyberSel [own source].

The rear fins can move at a frequency of up to 3 Hz within a range of about 80 degs outward, and up to 12 degs to the vehicle's inside axis of symmetry. The side fins can also move at up to 3 Hz over a range of ±45 degress. All electronic components, sensors, servos and a 7.2 V 10 Ah li-ion battery were enclosed in a sealed tube made of POM-C material. The servos with a 1:1 gearbox, responsible for the movement of the fins, are controlled via POLOLU-1353 miniMaestro servo controllers via RS232. The base station communicates via a WiFi network supported by a TP-Link TL-WR702N access point and a WIZNET Wiz-145SR server port providing four serial ports. The vehicle has an internal artificial buoyancy bladder, which is responsible for the vehicle's static depth adjustment or buoyancy control. The mini CyberSeal vehicle has been equipped with an OS-5000 digital compass from Ocean Server and an A-10 depth sensor from Wika with a measurement range of 0–1 bar. The sensors mentioned above make it possible to read current depth values and vehicle motion parameters, i.e., angle of an inclination concerning individual axes of the coordinate systems.

2.1.2. Mathematical Model

The model captures the underwater vehicle's rigid body dynamics, hydrostatics and hydrodynamic effects. Critical issues for the modelling and simulation of BUVs are model complexity, ease of implementation and accuracy of prediction. Therefore, for the mathematical description, some simplifications are adopted for the vehicle: it has three planes of symmetry, moves at low speed in a viscous fluid, and has six degrees of freedom. When analyzing the motion of an underwater vehicle, two reference systems are defined:

(1) a stationary $xyz$ coordinate system associated with the Earth,
(2) a moving $x_o y_o z_o$ coordinate system associated with the underwater vehicle.

The moving coordinate system is commonly called the "vehicle reference system", and its origin corresponds to the geometric center of the vehicle. The different axes of this coordinate system correspond to the following:

(1)　$x_o$—the longitudinal axis directed from the stern to the bow,
(2)　$y_o$—transverse axis directed to the starboard side,
(3)　$z_o$—vertical axis directed towards the bottom.

Changes in the position of the moving $x_o y_o z_o$ coordinate system are described relative to the adopted $xyz$ coordinate system associated with the Earth. Due to the low velocity of the vehicle, the acceleration of points on the Earth's surface due to its spin is neglected, and the $xyz$ system is considered stationary. It is suggested that angular and linear velocities be described in the reference system associated with the vehicle while the vehicle's orientation is described in a stationary coordinate system. The quantities describing the vehicle's movement are defined according to the SNAME notation in Table 1 .

$$\mathbf{M}\dot{v} + \mathbf{D}(v)v + \mathrm{g}(\eta) = \tau \tag{1}$$

where:

$v$—vector of linear and angular velocities, i.e., $v = [u, v, w, p, q, r]$;

$\eta$—vector of vehicle position and Euler angle coordinates in the stationary system;

$\mathbf{M}$—inertia matrix (equal to the sum of the rigid body mass matrix $\mathbf{M_{RB}}$ and the associated masses matrix $\mathbf{M_A}$);

$\mathbf{D}(v)$—hydrodynamic damping matrix;

$\mathrm{g}(\eta)$—matrix of restoring forces (gravity forces $P$ and buoyancy forces $B$);

$\tau$—vector of forces and moments acting on the vehicle.

**Table 1.** Notation used in describing the movement of underwater vehicles.

| Degrees of Freedom | Name of Movement | Forces and Moments | Angular and Linear Velocities | Position and Euler Angles |
|---|---|---|---|---|
| 1 | Movement in the direction of the $x_o$ axis | $X$ | $u$ | $x$ |
| 2 | Movement in the direction of the $y_o$ axis | $Y$ | $v$ | $y$ |
| 3 | Movement toward the $z_o$ axis | $Z$ | $w$ | $z$ |
| 4 | Rotation about the $x_o$ axis | $K$ | $p$ | $\phi$ |
| 5 | Rotation about the $y_o$ axis | $M$ | $q$ | $\theta$ |
| 6 | Rotation about the $z_o$ axis | $N$ | $r$ | $\psi$ |

Given the assumptions mentioned above, a nonlinear model of motion in six degrees of freedom is adopted for simulating the movement of the mini CyberSeal. The action of the vehicle is described by six differential equations, which, presented in matrix form, have the following format: The right side of Equation (1) represents the vector of forces and moments of force acting on the vehicle generated by the vehicle's propulsion system (2).

$$\tau = [X, Y, Z, K, M, N] \tag{2}$$

where:

$X, Y, Z$ —forces acting on the vehicle in the longitudinal, transverse and vertical symmetry axis, respectively;

$K, M, N$—moments of forces acting in relation to the longitudinal, transverse, and vertical symmetry axis, respectively.

The vector of forces and moments of forces generated by the wave propulsion can be calculated by considering the propulsion system set-up in each design. Figure 2 shows the mini CyberSeal propulsion model consisting of two counter-phased tail fins and two

independently controlled side fins. The thrust produced by each fin should be conveyed to the center of gravity *O* (Figure 2) using simple vector transformation formulas:

$$X = X_{tl} + X_{tp} + X_l + X_p \tag{3}$$

$$Y = Y_{tl} + T_{tp} \tag{4}$$

$$Z = Z_l + Z_p \tag{5}$$

$$K = 0 \tag{6}$$

$$M = M_l + M_p \tag{7}$$

$$N = N_{tl} - N_{tp} + N_l - N_p \tag{8}$$

where:

*tl*, *tp*, *l* and *p*—subscripts referring to the action of the left rear fin, right rear fin, left-side fin and right-side fin, respectively.

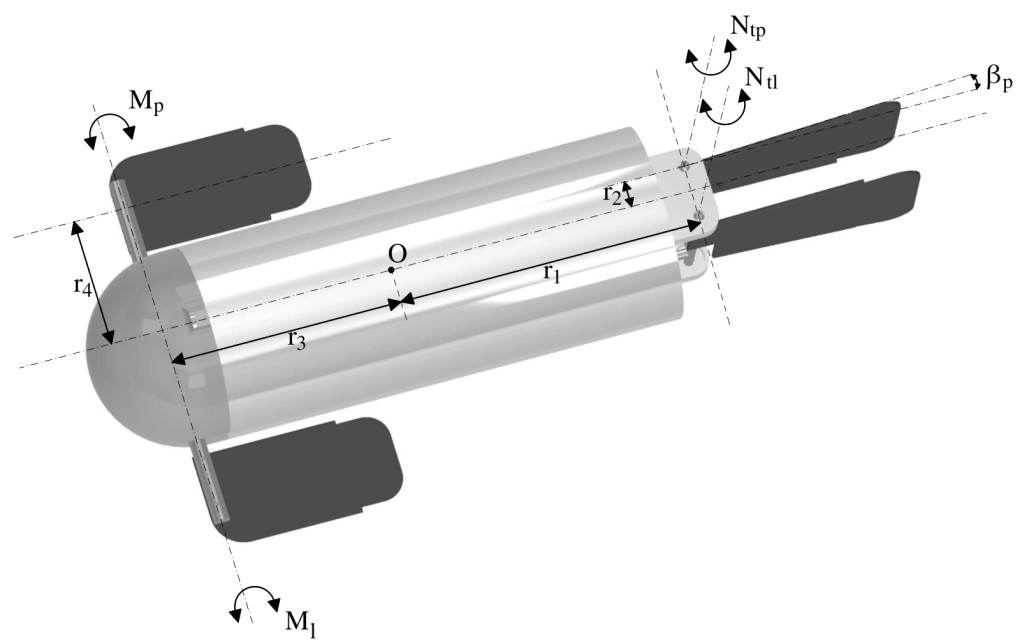

**Figure 2.** Mini CyberSeal propulsion model [own source].

The individual components of the vector, e.g., $X_{tl}$, $Y_{tl}$, $N_{tl}$ can be calculated using the position of these fins with respect to the centre of gravity according to the equations:

$$X_{tl} = \cos(\beta_l) * T_{tl} \tag{9}$$

$$X_{tp} = \cos(\beta_p) * T_{tp} \tag{10}$$

$$Y_{tl} = \sin(\beta_l) * T_{tl} \tag{11}$$

$$Y_{tp} = \sin(\beta_p) * T_{tp} \tag{12}$$

$$N_{tl} = r_2 * X_t l + r_1 * Y_{tl} \tag{13}$$

$$N_{tp} = r_2 * X_t p + r_1 * Y_t p \tag{14}$$

$$X_l = \cos(\alpha_l) * T_l \tag{15}$$

$$Z_l = \sin(\alpha_l) * T_l \tag{16}$$

$$M_l = r_3 * Z_l \tag{17}$$

$$N_l = r_4 * X_l \tag{18}$$

More information on the mathematical dependencies of the novel drive used is contained in the [21]. As shown in Figure 1 or Figure 3, the Mini CyberSeal has two side fins and two tail fins, which generate time-varying thrust.

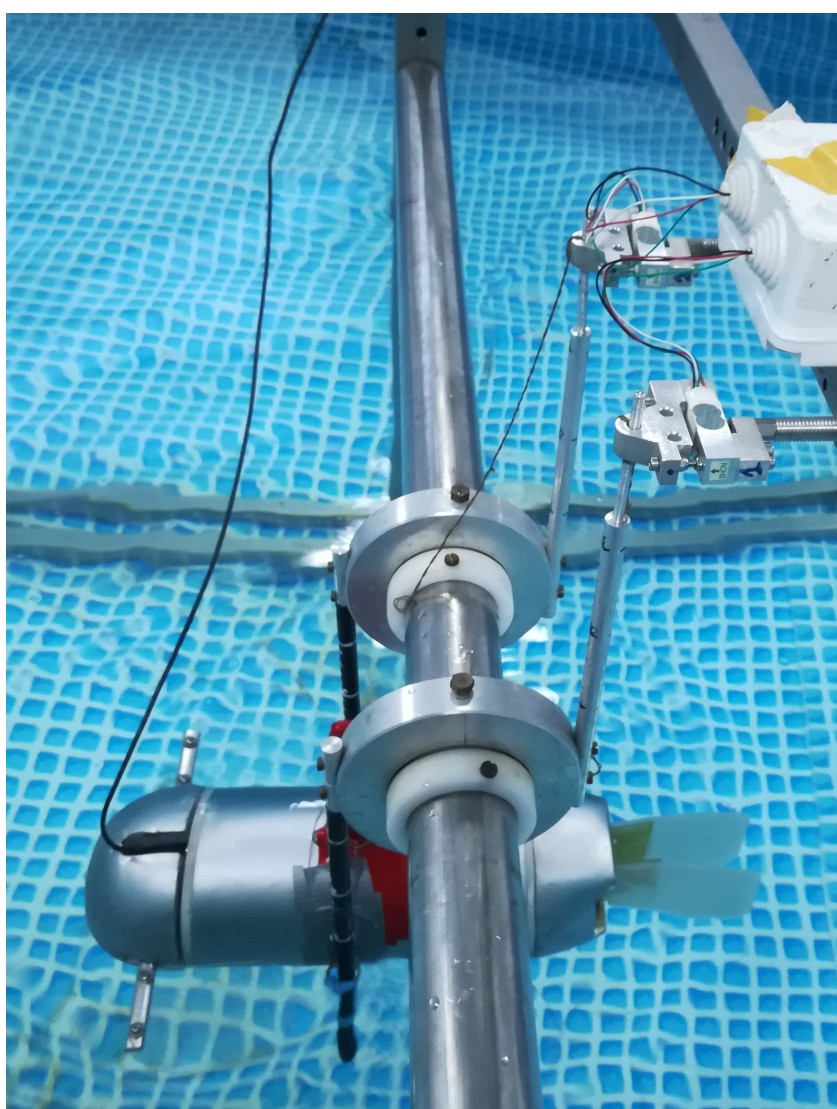

**Figure 3.** Mini CyberSeal thrust measurement stand [own source].

The value of the thrust $T_{tl}$, $T_{tp}$, $T_l$, $T_l$ depends on the control parameters, including the frequency and amplitude of the deflection of each fin. The thrust values generated by each fin for different frequencies and amplitudes were determined experimentally. Finally, the vehicle's speed is variable and dependent on the frequency of fin oscillations. The method of measuring the thrusts generated by the mini CyberSeal used in the mathematical equations is presented in the literature [20]. The thrust $T$ generated by the fin is the sum of two components:

$$T = T_{av} + T_{osc} \tag{19}$$

where:

$T_{av}$—constant thrust component at a specific fin oscillation frequency;

$T_{osc}$—variable component modelled by a sinusoidal wave with a specific amplitude (at a specific fin oscillation frequency).

At the same time, the test stand is shown in Figure 3. In addition, the right side of Equation (1) considers the effect of environmental disturbances such as wind, waves and sea currents, which significantly impact the BUV. The left side of Equation (1) describes the forces and moments of force caused by physical phenomena, such as rigid body inertia and the inertia of masses accompanying viscous fluid, the hydrodynamic drag exerted by water, and the balance of gravity and buoyancy forces. Using the mathematical relationships included in the literature [18], the matrix parameters describing the left side of Equation (1) can be calculated.

*2.2. Methods*

This subsection deals with the main part of the work concerning the selection of depth controllers' settings. The first subsection describes the controllers used, i.e., PID and SM. The following subsections describe the optimization methods and the fitness functions based on which the aforementioned methods realized optimization of used controllers settings.

2.2.1. Depth Controllers

Two classic controllers were used as depth controllers for the mini CyberSeal vehicle. The first one is about a PID controller that calculates the error value $e(k)$ as the difference between the set depth value and the value received from the depth sensor and applies a correction based on proportional, integral and derivative terms (denoted P, I, and D, respectively). Hence the name [22,23]. The action of the PID controller is described by the following formula presented in the discrete form:

$$u(k) = k_p e(k) + k_i \sum_{k=1}^{k_{max}} e(k) + k_d \Delta e(k) \tag{20}$$

where:

$u(k)$ is a control signal in $k$ step of simulation;

$e(k)$ is an error signal in $k$ step of simulation;

$\Delta e(k)$ is a change of error signals in $k$ step of simulation, i.e., $e(k) - e(k-1)$;

$k_p$, $k_i$ and $k_d$ are constant quantities called gain factors.

The second controller used is the sliding mode controller (SM) [24,25], where sliding mode control is achieved by controlling nonlinear systems, which changes the dynamics of a nonlinear system by applying a discontinuous control signal, which forces the system to "slide" along upstream of the expected behavior of the system. It is calculated using the following formulas, also presented in the discrete form:

$$s(k) = \frac{\lambda e(k) + \Delta e(k)}{\varphi} \tag{21}$$

$$\text{if} \quad |s(k)| > 1, \quad \text{then} \quad s(k) = sign(s(k)) \tag{22}$$

$$u(k) = k_s s(k) \tag{23}$$

where:

$u(k)$ is a control signal in $k$ step of simulation;

$s(k)$ is a normalized control signal in $k$ step of simulation;

$e(k)$ is an error signal in $k$ step of simulation;

$\Delta e(k)$ is a change of error signal in $k$ step of simulation, i.e., $e(k) - e(k-1)$;

$\lambda$, $\varphi$, $k_s$ is a constant settings of SM controller.

### 2.2.2. Optimization Methods

To tune controller settings, used the Global Optimization toolbox of the MATLAB environment [26,27]. The following three optimization methods were used: Genetic Algorithm (GA), Particle Swarm Optimization (PSO) and Pareto Simulation (PSA).

The genetic algorithm, first formalized as an optimization method by Holland, is a global optimization technique for multi-dimensional, nonlinear, and noisy problems and a stochastic search technique based on the mechanism of natural selection and natural genetics [28,29]. A genetic algorithm (GA) solves both constrained and unconstrained optimisation problems based on a natural selection process that mimics biological evolution. The algorithm repeatedly modifies a population of individual solutions. At each step, the genetic algorithm randomly selects individuals from the current population and uses them as parents to produce the children for the next generation. Changes are introduced into the offspring through mutation, crossover and other genetic operators. The procedure ends when satisfactory genotypes (a set of traits of an individual) are obtained, which are matched by phenotypes with a high fitness function (an individual from the population). Over successive generations, the population "evolves" toward an optimal solution. An initial population of 40 individuals was generated using a MATLAB random generator during optimization. Individuals in the current generation are estimated using one of the three fitness functions described in the following subsection. After calculating the fitness function, the reproduction algorithm creates children for the next generation. The following operators are used in reproduction: fitness rank scaling, Stochastic uniform selection function, Crossover fraction equal to 0.8, and Gaussian mutation function. The GA optimization was stopped when the maximum number of 100 generations was reached and/or when no change in the best fitness function value for new generations was detected during the next 50 steps [30].

The PSO algorithm is based on a simplified social model closely tied to swarming theory. It solves a problem by having a population of candidate solutions, here dubbed particles, and moving them around the search space according to simple mathematical formula over the particle's position and velocity. Each particle's movement is influenced by its local best-known position. Still, it is also guided toward the most notable positions in the search space, updated as better places are found by other particles [31]. A physical analogy might be a swarm of bees searching for food sources. In this analogy, each bee (referred to as a particle here) uses its memory and knowledge gained by the swarm to find the best available food sources. This is expected to move the swarm toward the best solutions. Based on the literature [32,33], the following PSO parameters were assumed: (1) MaxStallIterations (relative change in the value of the best objective function): 20, (2) MinNeighborsFraction (setting both the initial neighborhood size for each particle and the minimum neighborhood size): 1, (3) SwarmSize: 200. As in the case of GA, one of the critical problems is to properly define the fitness function to get the correct optimization rates. The fitness functions used were analogous to GA and are presented in the following subsection.

Pareto optimization (PSA) is a field of multi-objective decision-making that deals with mathematical optimization problems involving more than one objective function for simultaneous optimization. Multi-objective optimization has found application in many scientific areas, including engineering, economics and logistics, where optimal decisions must be made during trade-offs between two or more conflicting objectives. When we have several objective functions that we want to optimize simultaneously, these solvers find optimal trade-offs between competing objective functions. This method can also be applied to a single-objective problem. PSA uses pattern search on a set of points to iteratively search for non-dominated points. It should fulfil all constraints and linear constraints in each iteration. Theoretically, the algorithm converges to points near the true Pareto front [34]. In the algorithm used, in the beginning, the PSA creates an initial set of 200 randomly selected points and then checks whether these points are feasible concerning the bounds and linear constraints. If impossible, the algorithm projects the initial points into a linear subspace of linearly feasible points by solving a linear programming problem and removing

duplicate points. The PSA then divides the points into two sets named "archive" and "iterative". The archive set contains non-dominated points associated with a mesh size less than $10^{-6}$ and satisfying all constraints within $10^{-6}$. PSA checks the location of each point in the 'iterative' set. Success is achieved if the polled points yield at least one dominated point. PSA then extends the probing in successful directions multiple times, doubling the 6e grid to find a dominant point. If any non-dominated point is obtained, the grid size is halved. The algorithm stops when: (1) the mesh size exceeds the value (+Unity), (2) the fitness function decreases to the value (-Unity), (3) it reaches the maximum number of iterations equal to 400.

2.2.3. Fitness Function

It is necessary to determine proper fitness functions to obtain appropriate optimization results for the depth controller. For this paper, three functions commonly used in mathematics were formulated [35,36]. The first is Integral Absolute Error (ISA), the sum of the absolute values of the error signals $e(k)$ in all simulation steps. Its task is to select the controller parameters so that the error rate between the desired depth and the present depth value is as low as possible throughout the simulation period. The ISA in discrete form is shown in Equation (24).

$$f_{fit1} = \sum_{k=1}^{k_{max}} |e(k)| \tag{24}$$

The second fitness function is an Integral of Squared Error (ISE) in the discrete form Equation (25). The ISE integrates the square of the error over time. ISE will penalize for significant errors more than smaller ones (since the square of a large error will be much bigger). Control systems specified to minimize ISE will tend to eliminate significant errors quickly but will tolerate minor errors persisting for an extended period. Often this leads to fast responses but with considerable, low amplitude oscillation.

$$f_{fit2} = \sum_{k=1}^{k_{max}} e(k)^2 \tag{25}$$

The third proposed fitness function is based on combining two direct control quality indexes and is presented in the following Equation (26).

$$f_{fit3} = \sum_{i=1}^{i_{max}} t_r(i) + k_M \sum_{i=1}^{i_{max}} M_p(i) \tag{26}$$

It takes into consideration rising times $t_r$ in [s] and first overshoots $M_p$ in [rad] for all $i_{max}$ changes of the desired course. Because of the small value of $M_p$ in [rad] compared to $t_r$ in [s] additional gain factor of the sum of first overshoots was introduced $k_M = 25$.

## 3. Research Problem and Results

The research problem of this work is to find a solution for the most effective depth control of the mini CyberSeal vehicle. With a mathematical model of the vehicle, two types of controllers (PID, SM), three different methods for optimizing controller settings, and three fitness functions, an attempt was made to select a suitable controller and its parameters. All regulators, methods and functions should be examined in the tuning process and verified for each combination (Figure 4). The tuning process was carried out for three different immersion changes: (1) 0.2 [m]—shallow immersion, (2) 0.5 [m]—deep immersion, (3) and then two immersion changes—first 0.5 [m] then second 0.2 [m] depth change after 20 s of simulation. In contrast, the verification process was based on the tuning process for each combination of 25 randomly selected depths from 0 to 0.7 [m] (in 0.1 [m] increments). The average value of the fitness function was conducted from 25 verification tests.

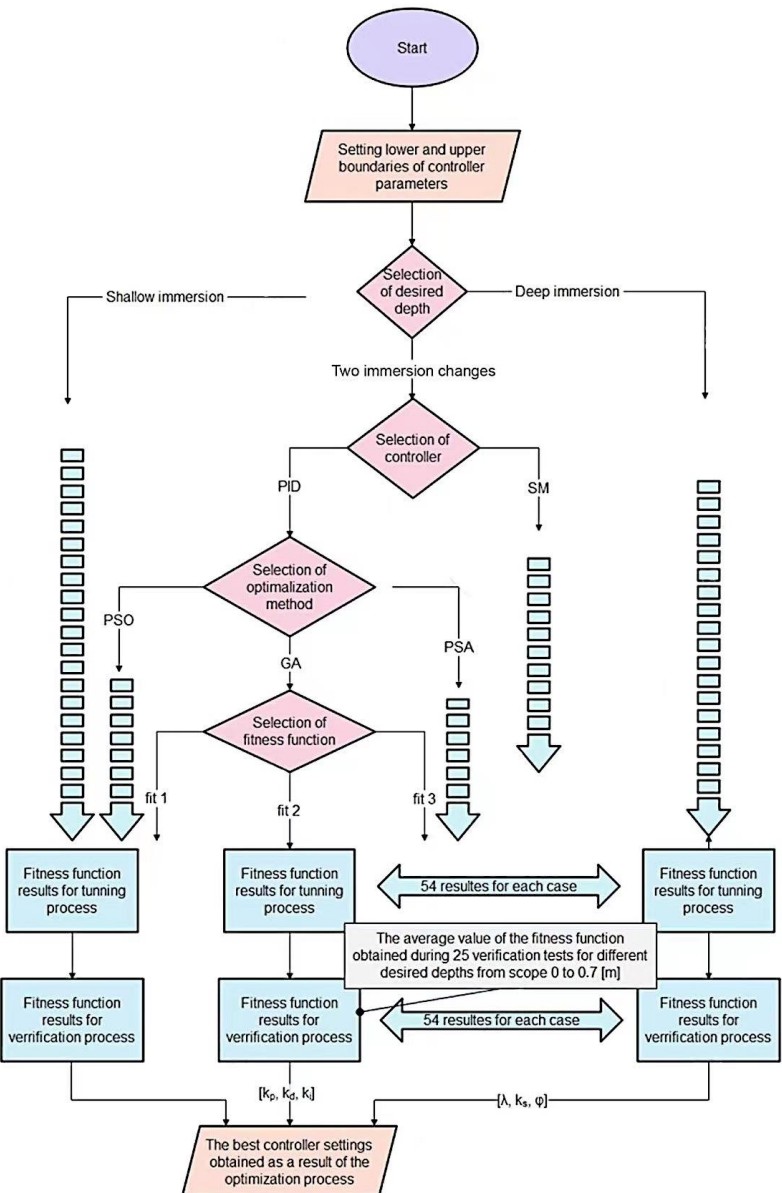

**Figure 4.** Flowchart optimization process to receive the best controllers settings.

*Results and Discussion*

The test results are shown in Tables 2–4 for the corresponding fitness functions 1–3. The table includes three different combinations of depth change for both the testing (T) and verification (V) processes. Also included in the table are combinations of controller type and its optimization method, e.g., SM-GA means SM controller was optimized by GA method, PID-PSA means PID controller was optimized by PSA method, etc. The initial study's objective was to find the optimal barriers for the controllers to avoid going into local minima. The research was carried out by an expert using a designed model of the CyberSeal, designed depth controllers and various tuning methods. As a result of the preliminary study, the upper and bottom barriers assume the following values: (a) for PID controller $[k_p, k_d, k_i]$—bottom barrier equals [200, 10,000, −1] and upper barrier equals [600, 40,000, 1], (b) for SM controller $[\lambda, k_s, \varphi]$—bottom barrier equals [−5, −100, −2] and upper barrier equals [5, 100, 2]. The main objective of the primary research was to compare two classic controllers, three methods optimized by fitness functions in response to changing the desired immersion value. The only quality control criterion optimized during tuning depth controllers was the minimized value of three different fitness functions. It means that all

phrases describing the "best results", "most effective", etc., which are used later in the article, refer to the smallest fitness function values.

Analyzing the results contained in Table 2 obtained for fitness function no. 1, the following conclusions can be drawn: (1) in the verification process, the best results for both controllers and all methods were obtained for a large change in immersion value, (2) in the tuning process, the best results for both controllers for all methods were received for a slight change in the immersion value, (3) PID-GA obtained the best result in the verification process. In contrast, SM-PSO and SM-PSA got the best result in the tuning process considering all of the depth changes, (4) optimization methods present similar efficiency for all of the desired depth changes, (5) the smallest values of the $f_{fit1}$ were obtained during tuning and verification for the SM controller than for the PID controller.

**Table 2.** Values of fitness function no. 1 for tuning (T) and verifying (V) BUV's depth controllers for three changes of desired immersion: shallow, deep and two following depth changes.

| Controller Type | Shallow (T) | Immersion (V) | Deep (T) | Immersion (V) | For Two (T) | Changes (V) |
|---|---|---|---|---|---|---|
| PID-GA | 30.5 | 48.6 | 73.3 | 55.5 | 128.6 | 54.2 |
| SM-GA | 29.2 | 88 | 71.3 | 54.3 | 126.2 | 96.2 |
| PID-PSO | 29.8 | 67.4 | 75.4 | 57.2 | 124.6 | 94.2 |
| SM-PSO | 30.1 | 92.1 | 65.6 | 55.2 | 116.5 | 95.1 |
| PID-PSA | 33.7 | 77,2 | 72.8 | 58.5 | 126.8 | 108.2 |
| SM-PSA | 32.1 | 116.1 | 67.6 | 96.8 | 120.1 | 99.7 |

Considering the results shown in Table 3 for fitness function no. 2, the following conclusions can be formulated: (1) in the verification process, the best results for both controllers and all methods were obtained for two changes in immersion value, (2) similar to earlier, the best results for both controllers for all methods in the tuning process were obtained for a slight change in the immersion value, (3) PID-GA obtained the best result in the verification process, while SM-PSO and SM-PSA obtained the best result in the tuning process taking into account all of the depth changes, (4) optimization methods present similar efficiency in the tuning process for all of the desired depth change.

**Table 3.** Values of fitness function no. 2 for tuning (T) and verifying (V) BUV's depth controllers for three changes of desired immersion: shallow, deep and two depths changes.

| Controller Type | Shallow (T) | Immersion (V) | Deep (T) | Immersion (V) | For Two (T) | Changes (V) |
|---|---|---|---|---|---|---|
| PID-GA | 11 | 36.4 | 84.2 | 53.2 | 141.8 | 30.8 |
| SM-GA | 11.2 | 75.1 | 85.6 | 81.2 | 167.8 | 80.1 |
| PID-PSO | 10.8 | 70.5 | 78.9 | 88.8 | 138.9 | 241.2 |
| SM-PSO | 11.4 | 165 | 72.2 | 78.2 | 129.8 | 77.2 |
| PID-PSA | 11.21 | 140.3 | 83.8 | 242 | 139.2 | 65.2 |
| SM-PSA | 11.2 | 199.8 | 76.8 | 358 | 134.4 | 92.6 |

Table 4 shows the results for fitness function no. 3. Analyzing its results, one can deduce: (1) as earlier, the best tuning process of both controllers was obtained for shallow immersion, while the best results for the verification process were received for two depth changes, (2) the smaller $f_{fit3}$ was obtained during tuning and verification process by PID controller than SM, (3) PID-PSO received the smallest value of $f_{fit3}$ in tuning for shallow immersion and verifying process for two depth changes, (4) optimization methods present similar efficiency for all of the desired depth change.

**Table 4.** Values of fitness function no. 3 for tuning (T) and verifying (V) BUV's depth controllers for three changes of desired immersion: shallow, deep and two following depths changes.

| Controller Type | Shallow (T) | Immersion (V) | Deep (T) | Immersion (V) | For Two (T) | Changes (V) |
|---|---|---|---|---|---|---|
| PID-GA | 8.21 | 14.1 | 9.26 | 15.9 | 20.1 | 9.11 |
| SM-GA | 8.9 | 22.41 | 14.76 | 24.2 | 26.2 | 15.6 |
| PID-PSO | 8.08 | 23.2 | 9.34 | 14.5 | 18.9 | 9.74 |
| SM-PSO | 9.01 | 24.6 | 11.45 | 24.31 | 27.6 | 30.41 |
| PID-PSA | 8.41 | 28.3 | 9.98 | 14.8 | 19 | 9.41 |
| SM-PSA | 9.5 | 25.5 | 12.32 | 15.81 | 31.2 | 15.81 |

Analyzing the above results in Tables 2–4, it can be concluded that the best results were obtained by PID-GA using objective function no. 3 for two depth changes. The result of $f_{fit3}$ indicates that the selected controller settings in the optimization process performed best in the verification process for random desired depth (for 25 random changes in depth from 0 to 0.7 m). From Figures 5–7 show the simulation results for the controller settings obtained for the PID-GA method, using the $f_{fit3}$ function and two depth changes. Figures 5–7 show, respectively, the results for each desired depth, i.e., 0.2 m, 0.5 m and two depth changes, first of 0.5 and then of 0.2. The simulation was carried out for 35 s. Considering the third case, the second desired depth signal occurred in 20 s of simulation. The timing was chosen so that the following depth changes occurred when there were no fluctuations in depth after the first change. The depth change was realized by the parallel swing of the side fins by the angle set by the control signal. In all the simulations, the oscillation frequency of the vehicle's side fins was constant and equal to 2 [Hz]. In the simulation, the side fin deflection angles were limited to 60 [deg] to avoid too excessive a vehicle trim angle. Each figure shows plots of the various parameters as a function of time: (1) graph of the dependence of the current vehicle depth on the desired, (2) values of the angular deflections of the side fins responsible for changing the depth of the vehicle based on the control signal along with the value of the vehicle trim (pitch), (3) error of depth over time. Analyzing the graphs presented, it can be noticed that selected controller gains are appropriate, and their selection is important for achieving good performance and stability of the system. The above graphs show that the object obtains satisfactory stability not only for the depths for which it was optimized (Figure 7) but also for other desired depths (Figures 5 and 6). The controller, in each case, reaches the required immersion quickly, and there is minimal overshoot. The controller's gains are not too large, so the system avoids exhibiting chattering, which is rapid switching between different control modes, which can lead to wear and tear on the actuators. The vehicle achieves stability within 12 s for all desired depths. To confirm the correctness of the adopted solution, i.e., tuning the controller for a specific value and then checking its settings for 25 random values, the simulation results for other controller settings are presented in Figure 8. For this figure, the controller settings were obtained for the PID-GA method, using the $f_{fit3}$ function and small depth changes. Comparing the simulation results for the same depth change (in Figures 5 and 8), but obtained for two different controller settings, it can be observed that the controller tuned for the specific depth reached the desired depth faster, i.e., 8 s (Figure 8). This is confirmed by the results in Table 4, wherein the tuning process for a small change in depth, the PID-GA obtained the value of the fit function 8.21, while for two changes in depth is 20.1. However, better results were obtained in the verification process for two depth changes for the entire depth spectrum (from 0 to 0.7 m). The value of $f_{fit3}$ for two depth changes was better by 50 per cent than for a small depth.

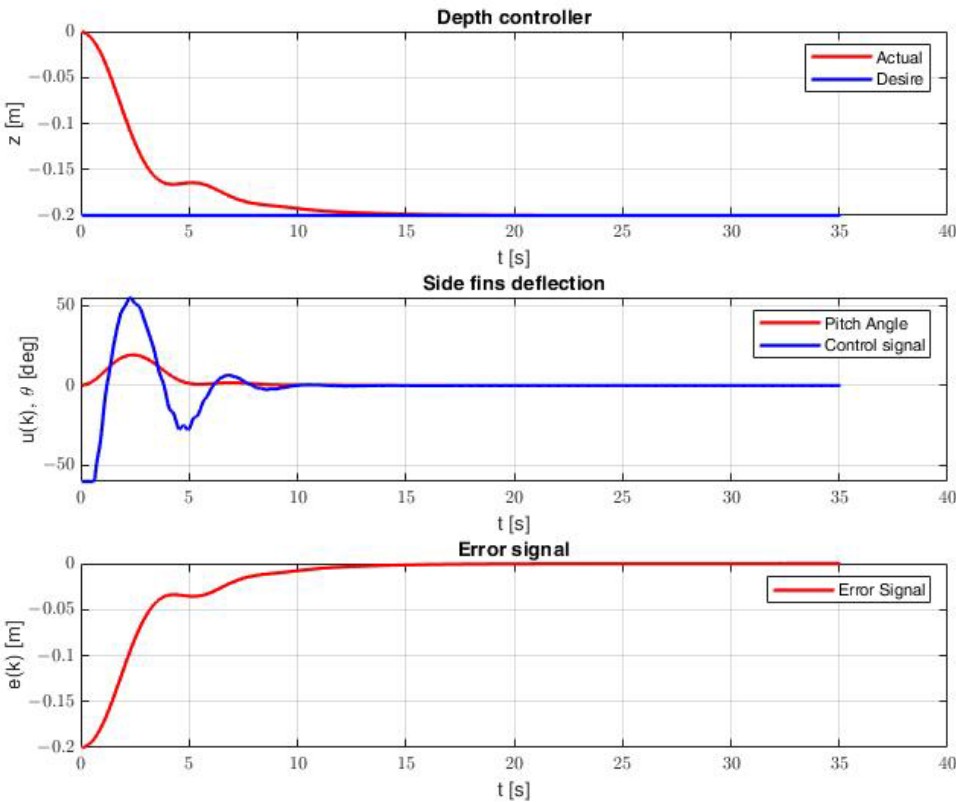

**Figure 5.** Changes of immersion and side fins deflection in time in response to desired depth: 0.2 [m] obtained for PID controller settings using GA method and fitness function no. 3 for two depth changes.

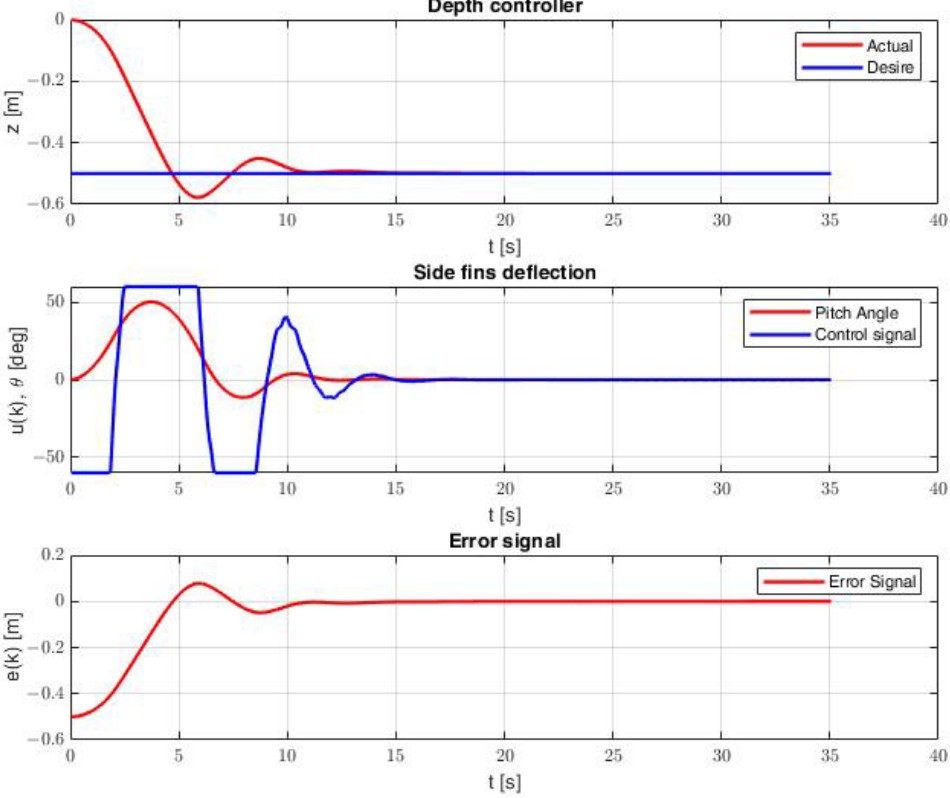

**Figure 6.** Changes of immersion and side fins deflection in time in response to desired depth: 0.5 [m] obtained for PID controller settings using GA method and fitness function no. 3 for two depth changes.

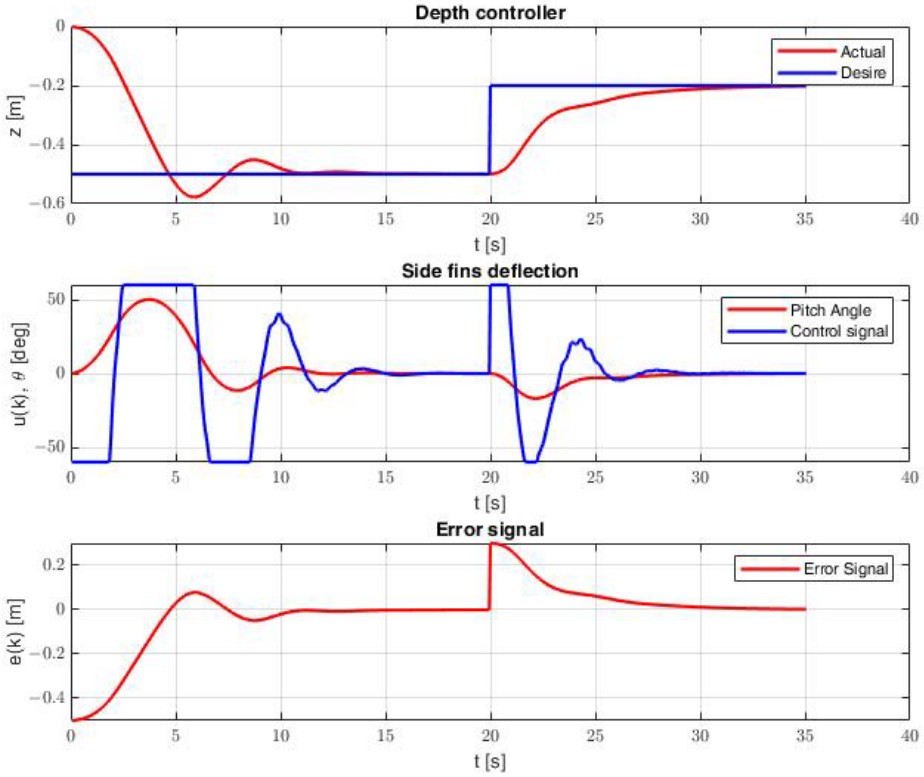

**Figure 7.** Changes of immersion and side fins deflection in time in response to subsequent desired depths: 0.5, 0.2 [m] obtained for PID controller settings using GA method and fitness function no. 3 for two depth changes.

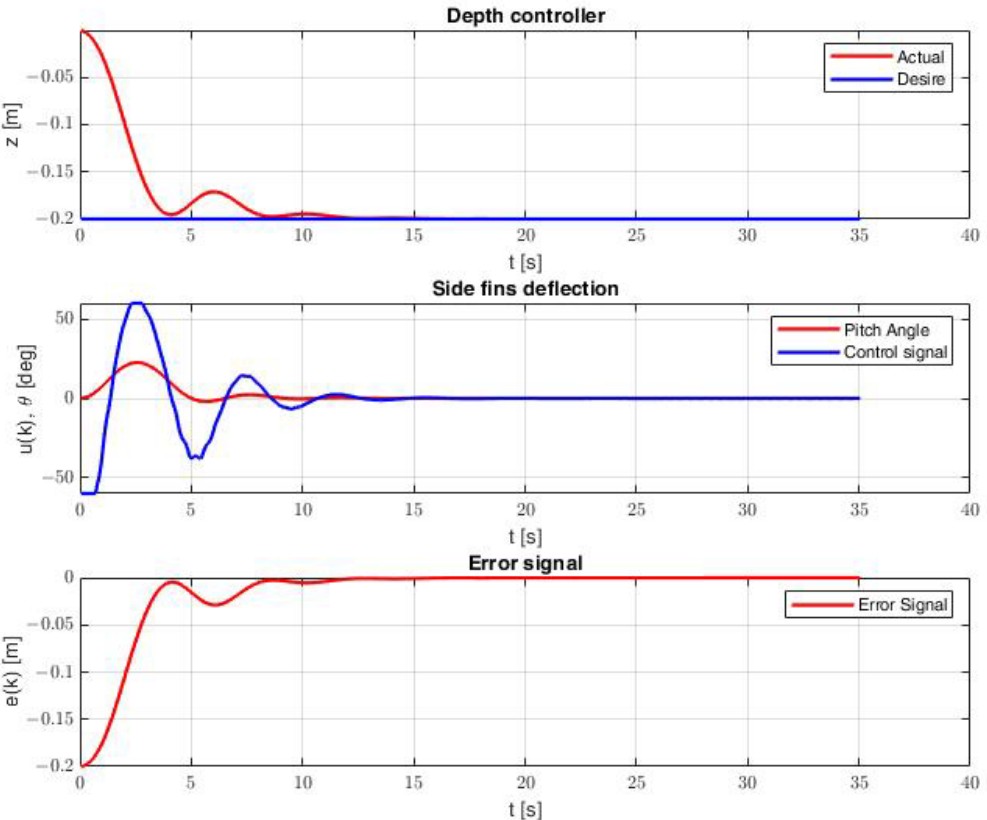

**Figure 8.** Changes of immersion and side fins deflection in time in response to desired depth: 0.2 [m] obtained for PID controller settings using GA method and fitness function no. 3 for small immersion.

## 4. Conclusions

This paper presents the tuning process of two classical depth controllers (PID, SM) of the biomimetic underwater vehicle with undulating propulsion. The tuning of the parameters of the controllers was carried out by applying three optimization methods and three fitness functions (GA, PSO, PSA), which provided a quality criterion. The optimization process was conducted for three different desired depth values, as shown in Figure 4.

Summarizing the research results obtained, several conclusions can be reached. The best results were not received with a slight change in immersion value. The results shown in Tables 2–4 confirm that despite the best values of the fitness function in the tuning process, the values for the verification process are several times higher. It can be recognized that the values of controller setting gains work efficiently only for small values. At the same time, they are not necessarily optimal for larger values used in the verification process. Also, the values of the fit function for different controller and optimization methods for the tuning and verification process are similar. Therefore, it is expected that the tuning of the controllers should be carried out for a larger desired depth or two or more changes of depth.

Furthermore, it can be deduced that, in most cases, better control quality was obtained for the PID controller than for the sliding controller. The differences can be seen in Table 4, where in all cases, both in the simulation and verification process, the fitness function obtained better values for the PID controller. The three optimization methods used to adjust the controller settings achieve comparable efficiency. Although the PSO and PSA methods achieved better tuning, the GA method achieved the best results in the verification process for each of the three different fitness functions.

Comparing the quality indicators, we can see that the ISE ($f_{fit2}$), which strongly penalizes any large deviation, obtained the worst results for a large change or two depth changes. In contrast, the best results were obtained for fitness function no. 3 ($f_{fit3}$) where used direct quality indicators, i.e., the rise time and the value of the first overshoots. It is because the fitness function obtained the most repetitive results for the tuning and verification process, and the values of fitness function no. 3 were the best compared to the other functions ($f_{fit1}$) and ($f_{fit2}$) using the classic indicators.

The presented results could be more comprehensive. An interesting issue would be further systematic testing of new multi-criteria indicators. In this way, it would be possible to use, for example, an LMS indicator or the like to limit sudden changes in the control signal and a more aggressive indicator such as ISE or ISE to increase the speed and accuracy of the process control. As part of future research, it is also planned to implement the applied regulators on a real object (mini CyberSeal) and verify them in natural conditions.

**Funding:** This research received no external funding

**Institutional Review Board Statement:** Not applicable.

**Informed Consent Statement:** Not applicable.

**Data Availability Statement:** Not applicable.

**Conflicts of Interest:** The author declares no conflict of interest.

## Abbreviations

The following abbreviations are used in this manuscript:

| | |
|---|---|
| AUV | Autonomous Underwater Vehicle |
| BUV | Biomimetic underwater vehicles |
| GA | Genetic algorithm |
| ISA | Integral Absolute Error |
| ISE | Integral of Squared Error |
| LMS | Least Median of Squares |

| PID | Proportional–integral–derivative controller |
| POM-C | Polyacetal (copolymer) |
| PSO | Particle Swarm Optimization |
| PSA | Pareto Simulation |
| ROV | Remote Operated Vehicle |
| SM | Sliding Mode controller |

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
