# Peer review of "Selection of the Depth Controller for the Biomimetic Underwater Vehicle"

_electronics, doi:10.3390/electronics12061469_

Round 1
Reviewer 1 Report
The paper is well written and easy to read. After a clear introduction, where all significant work related to the subject is covered, the paper provides a brief explanation of the vehicle’s dynamic model and control laws involved. The core of the work consists of optimizing the gains of the vehicle’s depth controllers (PID and sliding mode), based on different algorithms and different input conditions. Simulation results, representative of the controller performance are then presented.
I would encourage the author to add at least one sentence under each section title to describe how the section is split into the various subtitles (e.g. In section 2 Material method, write a sentence that introduces the reader on how that section is organized, including a forecast of the topics covered in the following subsections. The same for section 2.1 and 2.2, etc.)
The end of section 2.1.2 is very unfriendly for the reader, since it cites a lot of significant work without actually reporting anything. It would be nice to at least provide the equation for the drive configuration for the vehicle used in this study, so the reader does not have to dig it out from another paper.
While the effect of varying the PID gains is well known, the same cannot be said about the sliding mode gains because the sliding mode controller derivation (not shown) is usually based on Lyapunov functions/exponents that change depending on the system model. Thus, a brief explanation of how those terms affect the control output should be given.
Lastly, it would be significant to show how the controllers tuned via simulation performed when implemented on the vehicle, since usually sliding mode controllers are quite tedious to tune.
Author Response
Dear Reviewer,
Subject: Revision of the paper entitled “Selection of the depth controller for the biomimetic underwater vehicle”
Thank you for your comments and suggestions. I appreciate the time and details provided by you and have incorporated the suggested changes into the manuscript to the best of my ability. The manuscript has certainly benefited from these insightful revision suggestions. My responses are given in a point-by-point manner below.
After extensive revision, I hope that the manuscript is now suitable for publication and I look forward to hearing from you in due course.
Sincerely,
Michał Przybylski
Polish Naval Academy

Reviewer 2 Report
1- please, rewrite the sentences "The paper aims to select a depth controller for a novel biomimetic underwater vehicle ", such as "This paper ... "
2- please, rewrite the sentences "Two 1 different (PID and slide) controllers usually used for classical underwater vehicles were selected 2 for examination for a new vehicle with different undulating propulsion",
3- Care must be taken in writing the research. I advise the authors to send the paper to proofreading
4- the authors must adding more for related works in introduction section
5- It could be interesting to summarize the commented literature works in a table to have a clear comparison between all. This could also help precisely formulating the contribution of the paper with respect to previous works
6- please, replace the sentence " This paper describes the control object..." in the introduction to the contributions section
7- please adding a flowchart for proposed method
8- please support the conclusions section by results
9- The authors must compare the results of "Figure 3. Changes of immersion and side fins deflection in time in response to desired depth: 0.2 [m] produced by PID-GA obtained using fitness function no. 3. ", "Figure 4. Changes of immersion and side fins deflection in time in response to desired depth: 0.5 [m] produced by PID-GA obtained using fitness function no. 3. " and "Figure 5. Changes of immersion and side fins deflection in time in response to subsequent desired depths: 0.5, 0.2 [m] produced by PID-GA obtained using fitness function no. 3. " with other previous works
Author Response

(The authors gave the same response as above.)

Round 2
Reviewer 2 Report
no comments
Author Response
Dear Reviewer,
Subject: Revision of the paper entitled “Selection of the depth controller for the biomimetic underwater vehicle.”
Thank you for your comments and suggestions. I appreciate the time and details provided by you and have incorporated the suggested changes into the manuscript to the best of my ability. The manuscript has undoubtedly benefited from these insightful revision suggestions. English language editing also corrections of language mistakes, grammar and punctuation marked by blue fonts.
After extensive revision, I hope the manuscript is suitable for publication.
Sincerely,
Michał Przybylski
Polish Naval Academy
